

# Multi-horizon short-term load forecasting using hybrid of LSTM and modified split convolution

Irshad Ullah[1], Syed Muhammad Hasanat[1], Khursheed Aurangzeb[2], Musaed Alhussein[2], Muhammad Rizwan[2] and Muhammad Shahid Anwar[3]

[1] Electrical Engineering Kohat, University of Engineering & Technology Peshawar, Peshawar, KPK, Pakistan
[2] Department of Computer Engineering, King Saud University, Riyadh, Saudi Arabia
[3] Department of AI and Software, Gachon University Seongnam-si, Seongnam-si, South Korea

Corresponding authors
Irshad Ullah,
irshadullah@uetpeshawar.edu.pk
Muhammad Shahid Anwar,
shahidanwar786@gachon.ac.kr

## ABSTRACT

Precise short-term load forecasting (STLF) plays a crucial role in the smooth operation of power systems, future capacity planning, unit commitment, and demand response. However, due to its non-stationary and its dependency on multiple cyclic and non-cyclic calendric features and non-linear highly correlated metrological features, an accurate load forecasting with already existing techniques is challenging. To overcome this challenge, a novel hybrid technique based on long short-term memory (LSTM) and a modified split-convolution (SC) neural network (LSTM-SC) is proposed for single-step and multi-step STLF. The concatenating order of LSTM and SC in the proposed hybrid network provides an excellent capability of extraction of sequence-dependent features and other hierarchical spatial features. The model is evaluated by the Pakistan National Grid load dataset recorded by the National Transmission and Dispatch Company (NTDC). The load data is pre-processed and multiple other correlated features are incorporated into the data for performance enhancement. For generalization capability, the performance of LSTM-SC is evaluated on publicly available datasets of American Electric Power (AEP) and Independent System Operator New England (ISO-NE). The effect of temperature, a highly correlated input feature, on load forecasting is investigated either by removing the temperature or adding a Gaussian random noise into it. The performance evaluation in terms of RMSE, MAE, and MAPE of the proposed model on the NTDC dataset are 500.98, 372.62, and 3.72% for multi-step while 322.90, 244.22, and 2.38% for single-step load forecasting. The result shows that the proposed method has less forecasting error, strong generalization capability, and satisfactory performance on multi-horizon.

## INTRODUCTION

Electrical energy is a crucial commodity for the country's economic growth and is also essential to fulfilling daily activities in every walk of life. Therefore, its demand is rising exponentially worldwide due to its widespread use in both the production industrial sector

and non-production sectors. To fulfill this huge demand for electrical energy, and to reduce this huge pressure on the generation sector, some prosumers or independent power producers produce electrical energy by using environmentally friendly, carbon-free clean renewable energy sources such as wind and solar energy. They also share their energy with other smart microgrids. These renewable energy resources are highly intermittent in nature due to their dependency on weather conditions and other temporal features. Similarly, the load consumption behavior of consumers also depends on environmental and social factors. This increasing penetration of intermittent renewable generations and the complex nature of utility-customer behaviors add an extra level of system complexity and uncertainty in terms of electrical load demand. Therefore, accurate, precise load forecasting is highly challenging in dynamic situations due to its non-stationary behavior and its dependency on multiple calendric, meteorological, and other environmental and social factors.

To provide a reliable and sustainable power source to consumers at a consistent unit price for the best customer satisfaction and secure, economical, and reliable operation of the power system, accurate short-term load forecasting (STLF) is necessary. STLF also plays a vital role in electric load scheduling at different time horizons, future planning of power systems and energy resources, and tariff adjustment. It helps in the economical trading of electrical energy in the energy market for instance day ahead, intraday, and balancing markets. So, an inaccurate prediction leads to huge economic loss in-term of the operational cost of the power system, customer dissatisfaction cost, and poor management of energy sources (*Nti et al., 2020*). Therefore, electrical energy generation, transmission, and distribution networks governed by electric companies over the world need an accurate prediction of short-term load (STLF) for reliable and economical operations of power systems (*Khan et al., 2020*).

Load forecasting is broadly classified into three categories: short-term load forecasting (STLF), medium-term, and long-term load forecasting (LTLF). STLF predicts the load for the next hour to a few days and is useful for smooth operation and control of the power system. It is also helpful in energy management and load scheduling. On the other hand, medium-term load forecasting (MTLF) covers prediction from weeks to months while LTLF predicts from a few months to years. These two are important for long-term power system infrastructure planning (*Xia, Wang & McMenemy, 2010*; *Al Mamun et al., 2020*).

Electrical power can be viewed as time series data, and have all the specific attributes of time series data, such as seasonality, trend, and noise. It depends non-linearly on different environmental and social factors. In short, the power consumption data is sequence-dependent and non-stationary in nature. It can be analyzed more efficiently by extracting both its temporal features and spatial features. Therefore, for STLF many classical time series forecasting techniques and the recent machine learning and deep-learning techniques are used. Initially, many statistical methods such as auto-regressive integrated moving averages (ARIMA), and multiple linear regression are useful for predicting high-frequency time-series power consumption data. In contrast to statistical methods, artificial neural networks (ANN) are more powerful, especially in representing nonlinear behaviors

of the data, and performed better for STLF. However, deep ANN comes with the problem of over-fitting and slow convergence and also neglects the temporal or sequence-dependent features of load which is a key characteristic of time series data. To overcome this problem, first, recurrent neural networks (RNN) that emerged in the 1980s have the characteristic of extracting sequence-dependent features. However, this method has vanishing gradient problems which causes difficulty in the training process. To cover the problem of vanishing and exploding gradient, RNN architecture is modified and its new variant named long short term memory (LSTM) emerged in 1997. LSTM has received enormous attention for learning long-term dependencies in power consumption curve patterns using the special gated mechanism (*Sherstinsky, 2020*). Therefore, LSTM is widely used as a state-of-the-art method in the literature for STLF.

On the other hand, for extracting spatial features of time series load data such as trends, convolutional neural networks are used in literature which have special characteristics of extracting spatial patterns of data. To improve further the performance of STLF, hybrid networks are developed in the literature to handle both local trends and sequence-dependent features in an efficient way. In this regard, CNN-LSTM, CNN-BiLSTM is used in literature for STLF, in which CNN captures the local trends in load data pattern whereas the LSTM model captures the sequence-dependent pattern in electrical load which improve the prediction accuracy (*Rafi, Deeba & Hossain, 2021*).

However, in literature, the most recent hybrid methods CNN-LSTM are used in which stack of CNN layers preceded the LSTM layers. In this order, the CNN layers capture the spatial features and then pass through LSTM networks for temporal feature extraction which deteriorates the performance of hybrid models. To further enhance the performance of hybrid model, this study aims to exploit the potential, strengths, and weaknesses of different deep learning techniques on a real-time Pakistan National Grid data set. For this purpose, the article presents a novel hybrid technique based on long short-term memory (LSTM), and a modified split-convolution (SC) neural network (LSTM-SC) is proposed for single-step and multi-step STLF. In the proposed model, LSTM precedes and captures sequence-dependent features before extracting spatial features. Later CNN is also modified into a split parallel convolutional layer having different kernels size for extracting hierarchical spatial features. This specific order and modification improve the performance of STLF. The load data is also pre-processed and multiple other correlated features are incorporated into the data for performance enhancement. For generalization, the proposed model is also evaluated on publicly available datasets of American Electric Power (AEP) and Independent System Operator New England (ISO-NE). The performance evaluation in terms of RMSE, MAE, and MAPE of the proposed model is compared with the existing state-of-the-art models. The result reveals that the proposed method has better performance than existing methods and has strong generalization capability.

## Related work

The load data is non-stationary, weather-sensitive, and depends upon many temporal correlated features such as festivals, holidays, and other calendric features. This makes the STLF very challenging and demanding in the presence of non-linear electrical load

consumption. In this regard, various research has been done in the literature. The traditional statistical methods for time series data forecasting such as multiple linear regression (*Lee & Ko, 2011*; *Amral, Ozveren & King, 2007*) exponential smoothing (*Christiaanse, 1971*; *Taylor, 2003*) and auto-regressive integrated moving average (ARIMA) (*Lee & Ko, 2011*) are used for STLF. Similarly, machine learning method such as support vector regressions (SVR) (*Chen et al., 2017*; *Che & Wang, 2014*; *Li et al., 2007*) is used for performance enhancement. These statistical and machine learning methods performed well for time series data prediction but do not yield high accuracy in large amounts of uncertain and non-linear electrical load data.

On the other hand for a large amount of non-linear electrical load, deep learning techniques are widely used in literature for achieving better results. In *Ekonomou, Christodoulou & Mladenov (2016)*, *Sahay & Tripathi (2013)* authors used an ANN-based method along with the wavelet signal processing techniques for performance enhancement. However, the issue associated with ANN is the poor generalization due to trapping in local minima, which causes over-fitting and slow convergence. Another problem with ANN is that it neglects the intrinsic characteristics existing in the time series data. To resolve this issue, the long short-term memory (LSTM) introduced by Hochreiter and Schmidhuber (*Yu et al., 2017*) has received enormous attention in the realm of sequence learning. Therefore, LSTM is widely used in the literature for STLF. *Kong et al. (2017)* used two layers LSTM network, trained on 69 houses' load data, and compared the results with several other existing state-of-the-art models. In *Ageng, Huang & Cheng (2021)*, LSTM-DP combines data preparation with LSTM, the author pre-processes building load data and then extracts the pattern by using stack LSTM layer for next-hour load predictions. *Son et al. (2022)* makes a stack of LSTM layers, in which the first layer is a bidirectional LSTM layer, followed by two LSTM layers. The initial bidirectional layer extracts temporal features from the energy consumption sequence in both forward and backward directions. Similarly, in *Marino, Amarasinghe & Manic (2016)*, two LSTM networks: standard LSTM and LSTM-based sequence to sequence (S2S) architecture are used for load forecasting. In S2S both the encoder and decoder are developed by using LSTM. The input to the encoder is the date and time information along with the load and the load is delayed by one step. On the other hand, the input to the decoder is the date and time of the corresponding load which is to be forecasted. The result shows that S2S works better than standard LSTM. In *Ijaz et al. (2022)*, the combination of ANN and LSTM is used for short-term electrical load forecasting. This method is trained and tested on the Malaysian electric supply company dataset. The methods incorporated different weather and temporal features with load data such as humidity, holidays and date-time features. On the other hand, *Shao & Kim (2020)* used three parallel channels of LSTM and K-means classifier for multi-step STLF.

In addition to the above techniques, convolutional neural network (CNN) which has an excellent ability to capture the spatial features is excessively applied for STLF. In *Cho et al. (2014)*, data is reshaped into an image and then applied CNN for STLF. In *Kuo & Huang (2018)*, Deep Energy, a powerful model, based on a convolutional neural network and CNN-based bagging approach is used for predicting the load. It is also pertinent to

mention that CNN is one of the most popular algorithms of deep learning and developed a lot. Many articles in the literature are inspired from advance CNN architectures. For instance, in *Chen et al. (2018)* historical load and temperature with intelligent past days data framing are applied to a complex architecture based on the DenseNet (*Huang et al., 2017*) and ResNet (*He et al., 2016*) *i.e.*, skip connections for STLF.

LSTM network has the ability to extract the sequence pattern information from data and is used to exploit short-term and long-term dependencies. On the other hand, CNN is used to extract valuable spatial features. In addition, CNN may filter out the noise from the input data which eventually enhances the deep learning model performance. So, the integration of CNN and LSTM as a hybrid model, keeping the synergy of both for STLF, for performance enhancement, recently emerges its use in literature. In *Rafi, Deeba & Hossain (2021)* and *Alhussein, Aurangzeb & Haider (2020)*, a hybrid model (CNN-LSTM) composed of CNN layers cascaded in series with LSTM layers followed by an output dense layer. In *Rafi, Deeba & Hossain (2021)*, only load data is used as input to the network for feature extraction. In contrast, in *Alhussein, Aurangzeb & Haider (2020)* other three well-known co-related calendric features *i.e.*, an hour of the day, a day of the week, and a holiday indicator are incorporated with load data for better prediction. The model (*Alhussein, Aurangzeb & Haider, 2020*) outperforms the famous state-of-the-art forecasting models (*Kong et al., 2017*). Similarly, in *Sajjad et al. (2020)* a stack of two layers of CNN is followed by two layers of GRU while in *Ullah et al. (2019)*, a two-layer CNN is followed by Multi-layer Bi-Directional LSTM (M-BDLSTM) layer. Similarly, *Chen et al. (2023)*, *Hussain et al. (2022)*, used hybrid models composed of CNN and RNN variants for load forecasting. However, in *Chen et al. (2023)*, the authors used CNN followed by four ResNet modules. These existing hybrid models outperform the previous state-of-the-art models.

However, these articles use a stack of CNN layers, which precede the LSTM layers. In this order of hybrid network, the time-series data is first passed through CNN which extracts spatial features before extracting intrinsic time-dependent features which degrade the overall performance of the hybrid network. Although, it extracts some valuable features and enhances the results as compared to the individual LSTM. For extracting non-linear features using non-linear deep neural network modules, the arrangement of sub-modules in the network does matter. Therefore, it is logical to anticipate that by reversing the order in a hybrid model and keeping care of the temporal nature of load data improves the performance. Hence, keeping LSTM before CNN captures the sequence-dependent features of the load data in an efficient way and further passing it through CNN reduces forecasting errors. In addition, the stack CNN is further modified by a special split convolution (SC) network that extracts both local and global features. This hybrid network of LSTM with modified SC has not been yet implemented. To this end, this article intends to make the following contributions.

- A hybrid model of LSTM, modified SC is proposed for Multi-horizon short-term electrical load forecasting. This hybrid network is designed in such a way that the synergy of LSTM and SC is exploited.

**Table 1 The description of dataset.**

| Parameter | Value |
| --- | --- |
| Number of samples before processing | 46,606 |
| Number of the sample with NaN | 2 |
| Minimum before processing | 473 |
| Maximum before processing | 24,786 |
| Missing values | 120 |
| Date error | 53 |
| Outliers | 10 |
| Number of samples after processing | 46,728 |
| Minimum after processing | 1,648 |
| Maximum after processing | 22,696 |

- A unique strategy "split-transform-merge" is adopted for the CNN network instead of a stack of layers. This strategy is powerful for extracting embedded hierarchical both local and global features from the data pipeline. Moreover, the complicated network parameters—the filter size, number, activation, *etc.*—are uniquely tailored for energy consumption data.

- The proposed model is developed and evaluated on the Pakistan power system (PPS) load consumption data. The PPS load consumption data is not strictly following a consistent pattern and there are many perturbing and uncertain parameters for the load variation in Pakistan. Furthermore, the model's generalization is assessed by evaluating the model performance on two publicly available datasets: AEP and ISO-NE.

- The model is used to forecast both single-step and multi-step. In single-step, the model predicts the next hour's load, while in multi-step the model predicts the next 12 and 24 h ahead load consumption.

The rest of the article is organized as follows. 'Methodology' presents the methodology which covers exploratory data analysis (EDA), data set pre-processing, data framing, and model architecture. Then, 'Result and Discussion' comprises results and discussion. Next, 'Limitations and future work' contains future work, and finally, the 'Conclusion' concludes the article.

# METHODOLOGY

## Explanatory data analysis

In this section one of the three datasets is pre-processed, analyzed, and discussed in detail. Similarly, the other two datasets are processed. The hourly load profile data of Pakistan National grids from $1^{st}$ January 2015 to $30^{th}$ April 2020, recorded by NTDC is used. Some information regarding dataset is given in Table 1. Further sections describe more about the process of refilling the missing value, outliers handling, correlated features classification and correlation, seasonality, and non-stationary of electrical load consumption data behaviors in detail.

**Table 2 Missing values details.**

| Missing data points | Location |
|---|---|
| Complete day | 29 February 2016 |
| 24 consecutive data points | 16 points on $5^{th}$ May and eight points on $6^{th}$ June in 2015, 16, 17 and 18 |
| Two consecutive | 14 points |
| Single point | 40 points |

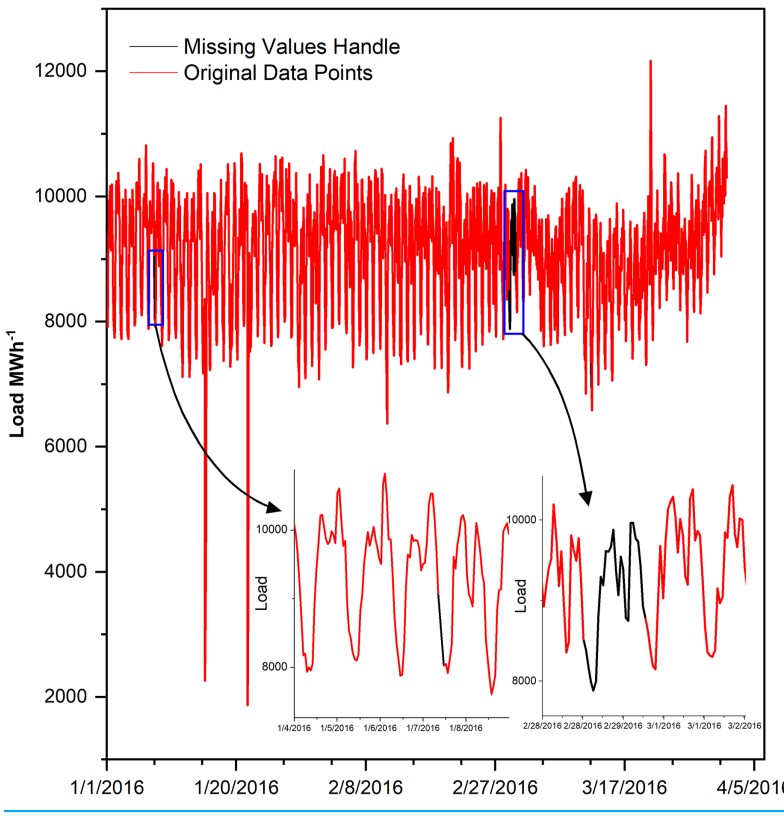

**Figure 1 Missing values in the dataset.**

### Data cleaning

*Missing values handling*

Missing values cause lousy forecasting, if not handled (*Wang et al., 2018*). Therefore, after analyzing data comprehensively, it is found that the data is missing in two ways, either many data points are missing consecutively or one or few points at different locations, the detail is given in Table 2. The one or few missing points are filled by interpolating with parameter time while many consecutive missing points are filled by averaging a load of last and next week, in such a way that the same hour of the missing day is filled by the average of the same hour load of last and coming week as shown in Fig. 1.
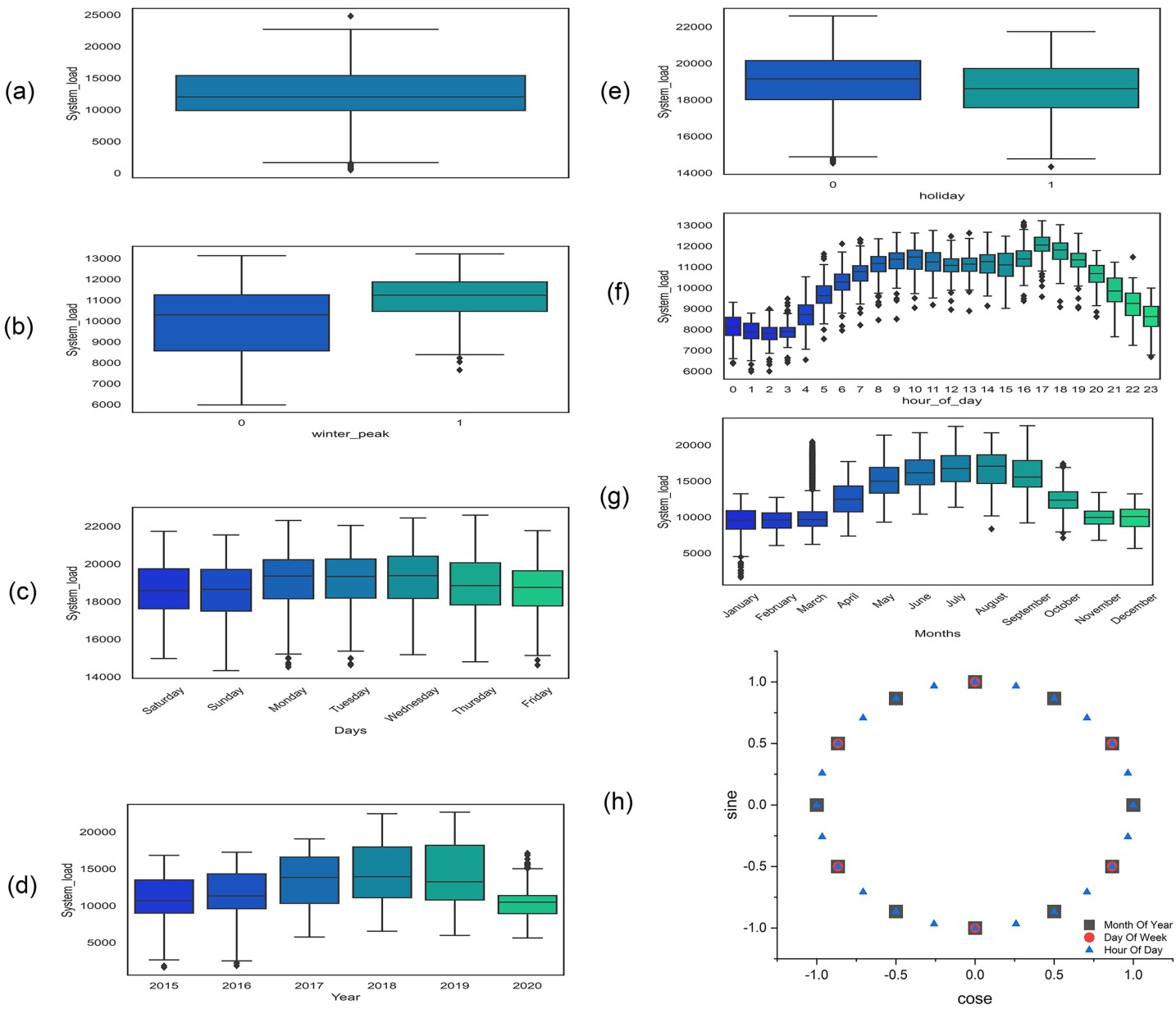

**Figure 2** Box plot at different steps of exploratory data analysis (A) outliers in dataset (B) peak hours: one for peak and 0 for non-peak hours (C) days of the week (D) yearly load (E) one for holiday and 0 for the working day (F) 24 h load consumption (G) power consumption of different month of the year (H) cyclic feature encoding of the month of the year, days of the week, and hours of the day.

*Outliers handling*

Similarly, the outliers are sudden high or low values at certain points which make load forecasting very challenging and difficult (*Shao & Kim, 2020*) and are identified by using the interquartile range (IQR) method (*Seo, 2006*) as shown in Fig. 2A. These are first removed from the data and then handled just like missing values. The other repeated values are also removed and handled like missing values.

**Table 3 Peak hours and off-peak hours.**

| Season | Peak hours | Off-peak timing |
| --- | --- | --- |
| Winter (Dec to Feb inclusive) | 5 to 9 pm | The remaining 20 h |
| Spring (March to May inclusive) | 6 to 10 pm | The remaining 20 h |
| Summer (June to Aug inclusive) | 7 to 11 pm | The remaining 20 h |
| Autumn (Sept to Nov inclusive) | 6 to 10 pm | The remaining 20 h |

**Table 4 Spearman correlation with features.**

| Feature | Spearman correlation |
| --- | --- |
| Month of year | 0.227 |
| Day of week | −0.018 |
| Hour of day | 0.148 |
| Holiday | −0.012 |
| Winter peak | −0.134 |
| Summer pear | 0.255 |
| Spring peak | 0.023 |
| Autumn peak | 0.053 |

### Co-related features

After data cleaning, the load is shown in Fig. 2D which shows a clear increasing trend and variation over the years which makes it highly non-stationary. The load consumption also vary with the different season over the year as shown in Fig. 2G, each and every season have different peak and off-peak hours in each day as shown in Figs. 2B and 2F. The complete detail of peak hours in different seasons is given in Table 3 (*National Transmission & Despatch Company (NTDC), 2019*). Similarly, working days, weekend days, and other public holidays also affect the load consumption as shown in Figs. 2C and 2E. Moreover, the correlation coefficient of all of the above features is tabulated in Table 4.

## Data set prepossessing

Table 4 shows that the load data variation has a correlation with other categorical features. So, the incorporation of these correlated features along with load data increases the forecasting accuracy. Hence, the input vector has both numerical load data (L) and categorical data and the later is further classified into cyclic categorical data which repeat itself after certain duration such as a month of the year (M), a day of the week (D) and an hour of the day (H) while non-cyclic categorical data are Holidays ($H_o$) either national holidays or weekends. Note that both public holidays and weekends are treated as a single binary variable. In the future, it may be split into cyclic and non-cyclic. Similarly, peaks hours depend on seasons: winter peak ($W_p$), spring peak ($S_p$), summer peak ($SU_p$) and autumn peak ($A_p$). Since this vector is used as input vector to deep learning models which

**Table 5 Detail of the input vector.**

| Data | Description | Encoding technique | Shape |
|---|---|---|---|
| Numerical | Load (L) | Min-max normalization | 1 |
| Cyclic | Hour of the day (H) | Trigonometric Transform | 2 |
| | Days of week (D) | | 2 |
| | Month of year (M) | | 2 |
| Non-cyclic | Holidays (Ho) | One-hot encoding | 2 |
| | Winter peak-hr ($W_p$) | | 2 |
| | Spring peak-hr ($S_p$) | | 2 |
| | Summer peak-hr ($SU_p$) | | 2 |
| | Autumn peak-hr ($A_p$) | | 2 |

are sensitive to data scaling, the load data and all the other categorical features are transformed to the ranges 0 and 1.

### Numerical data
The numerical load is normalized between 0 and 1 inclusive using min-max normalization as per Eq. (1) (*Farsi et al., 2021*).

$$L_{T_{Norm}} = \frac{L_t - L_{min}}{L_{max} - L_{min}} \tag{1}$$

where the $L_{T_{Norm}}$ is the normalized values between 0 and 1 while $L_t$ is the original value at time t and $L_{max}$ and $L_{min}$ are the maximum and minimum values of the load respectively.

### Categorical data
The categorical data are mostly calendric features, having both cyclic and non-cyclic behavior and encoded accordingly.

*Cyclic features*
The cyclical features are encoded *via* trigonometric transform, in which each sample is represented as (x, y), coordinates of a unit circle. The motivation behind this transformation is that the initial value of the cyclical feature is next to the final value. For instance, January is near to December in order and away from June, which is exactly captured by this transformation shown in Fig. 2H.

*Non-cyclic features*
The non-cyclical features are encoded by one hot encoding.

### Input matrix
All the processed input features are combined in a single matrix. The detail of the input vector is shown in Table 5. All these vectors are concatenated into a single vector

$$X = \{L, H, D, M, H_o, W_p, S_p, SU_p, A_p\} \tag{2}$$

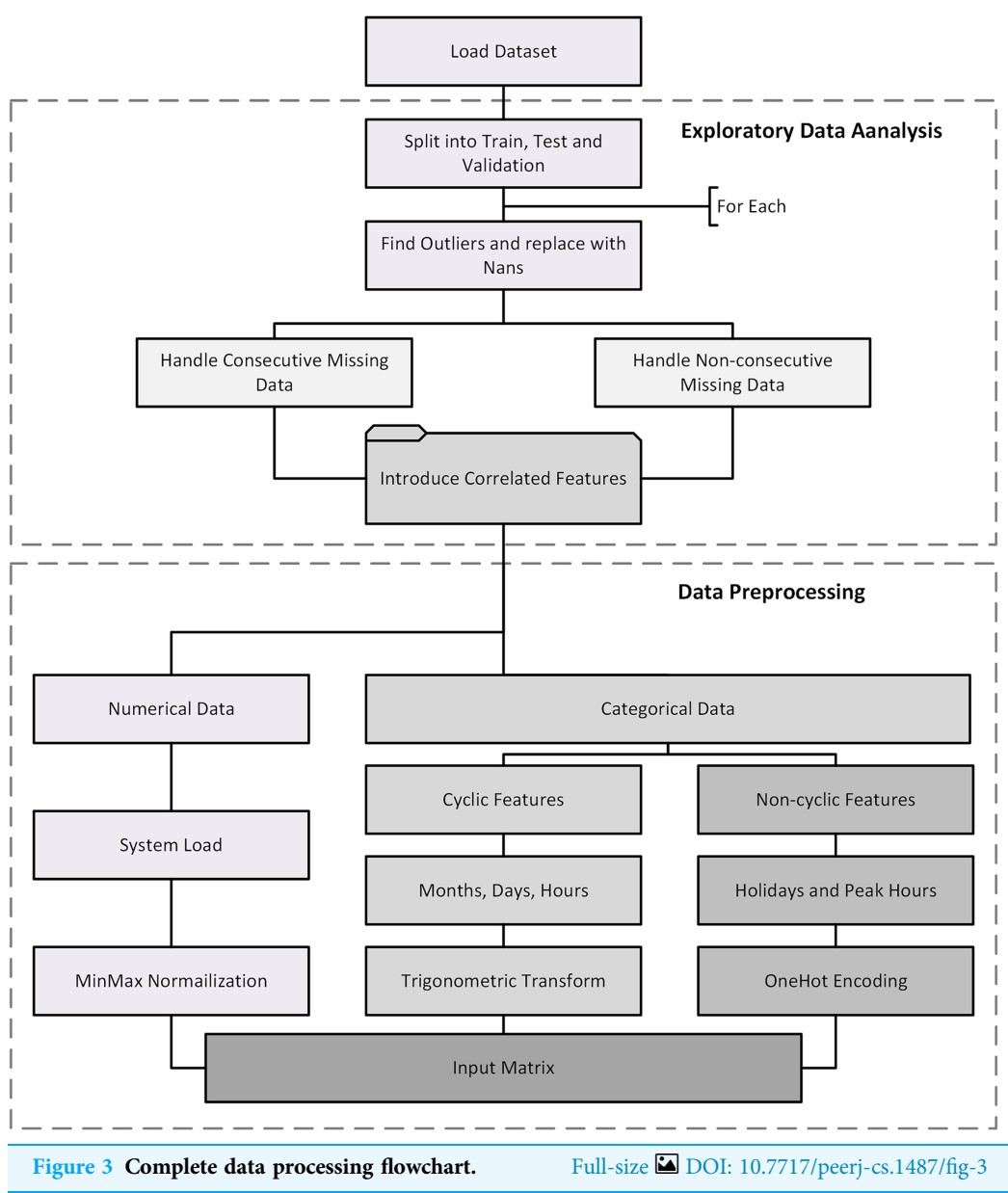

**Figure 3  Complete data processing flowchart.**

which makes a vector of size $17 \times 1$. The complete flow diagram of preprocessing is shown in Fig. 3.

## Data framing for single-step and multi-step forecasting

The concatenated data are a series of data points. To input this data into the LSTM, CNN, or hybrid models it must be in the shape (sample, time step, features) and corresponding labels. The labels depend on the task at hand. In this work, two types of tasks are a consideration, one is a single step ahead and another is multi-step. For the load data generation in proper framing for single-step ahead Algorithm 1 is used and for multi-step Algorithm 2 is used.

**Algorithm 1** Pseudo-code of load data generator for single step.

**Ensure:** $T_{set}$ < length of $D_{set}$

1: $D_{set}$: numpy array

2: $T_{set}$: Time-step

3: $I_{fcst}$: Index of Forecasting Column in $D_{set}$

4: Create two list A,B

5: **for** i < length of $D_{set}$-1 **do**

6:     $I_x = I + T_{step}$

7:     **if** $I_x$ > length of $D_{set}$-1 **then**

8:         Break

9:     **end if**

10:     Temp = data in $D_{set}$ from i to $I_x$

11:     Append Temp in A

12:     Temp = Data in $D_{set}$ at $I_x$, $I_{forecast}$

13:     Append Temp in B

14: **end for**

15: **return** Array of A,B

 

**Algorithm 2** Pseudo-code of load data generator for multi step.

**Ensure:** $T_{set}$ < length of $D_{set}$

1: $D_{set}$: numpy array

2: $T_{set}$: Time-step

3: $I_{fcst}$: Index of Forecasting Column in $D_{set}$

4: $F_{len}$: Forecast Duration

5: Create two list A,B

6: **for** i < length of $D_{set}$-1 **do**

7:     $I_x = I + T_{step}$

8:     **if** $I_x + F_{len}$ > length of $D_{set}$-1 **then**

9:         Break

10:     **end if**

11:     Temp = data in $D_{set}$ from i to $I^x$

12:     Append Temp in A

13:     array = $I_{fcst}$ of $D_{set}$

14:     Temp = Data in array from $I_x$ to $I_x + F_{len}$

15:     Append Temp in B

16: **end for**

17: **return** Array of A,B

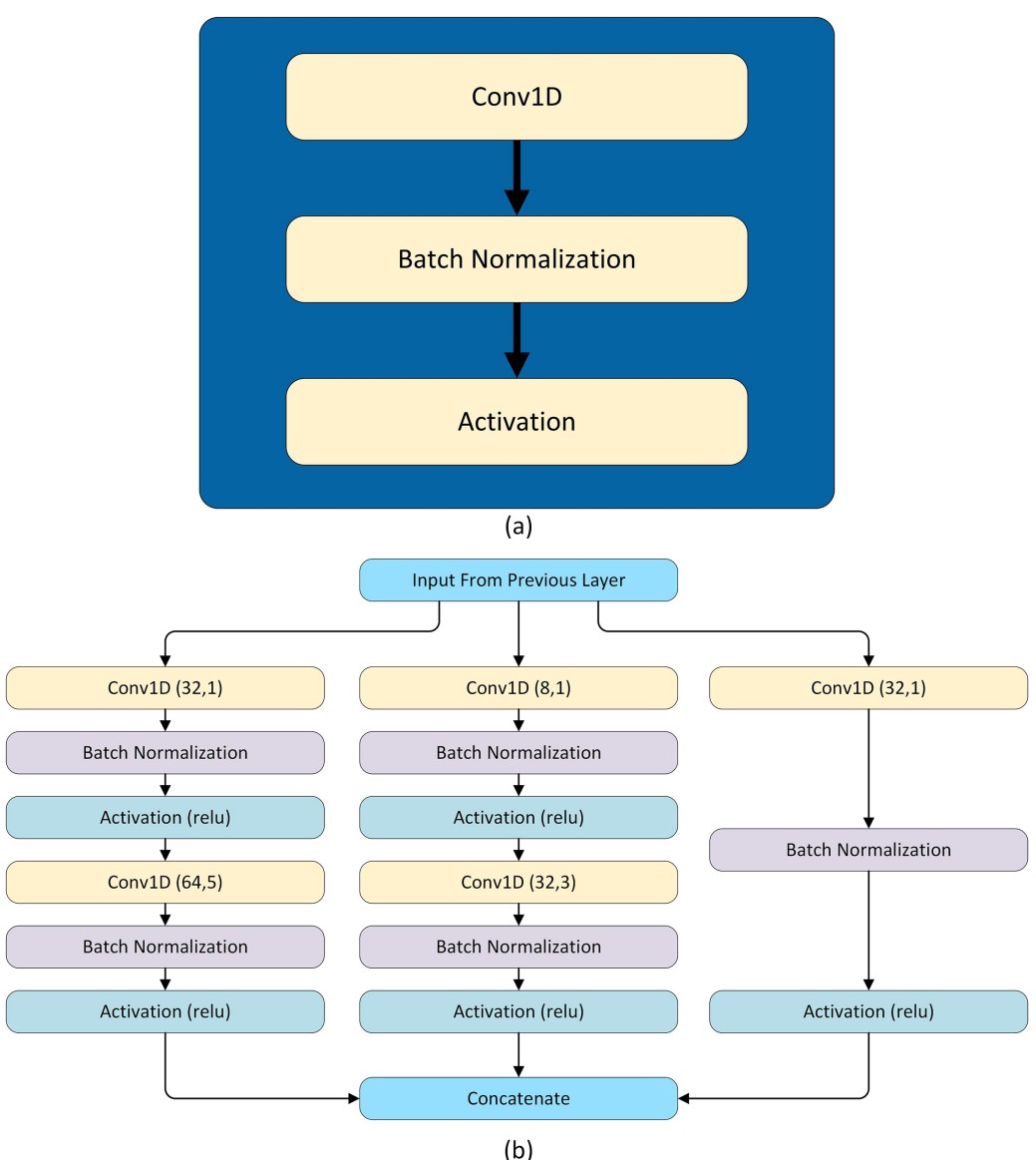

**Figure 4  Architectural ingredients of the proposed model (A) convolutional block (B) SC module.**

## Model architecture

The model is tailored for forecasting the load. The hybrid model is composed of two LSTM layers, two modified SC modules, and three skip connections. Each LSTM layer has 48 units and these LSTM layers extract temporal features from the load. These features are passed through two modified SC modules. The SC is inspired by *Szegedy et al. (2017*, *2015*, *2016)*. Each SC module has three parallel paths for data flowing and each of these paths has a different number of CNN layers having different filter sizes to extract both global and local features. In addition, the three skip connections are made in such a way that two out of three are taken from the output of the second LSTM layer and concatenated with the output of first and second SC modules. However, this introduces unequal representation to
Table 6  SC module hyperparameters FP: First path, SP: Second path, TP: Third path.

| Model | Inception | 1 × 1 (FP) | 1 × 1 (SP) | 3 × 3 (SP) | 1 × 1 (TP) | 5 × 5 (TP) |
|---|---|---|---|---|---|---|
| Proposed hybrid | Inception 1 | 32 | 32 | 64 | 8 | 32 |
| | Inception 2 | 64 | 48 | 64 | 16 | 64 |

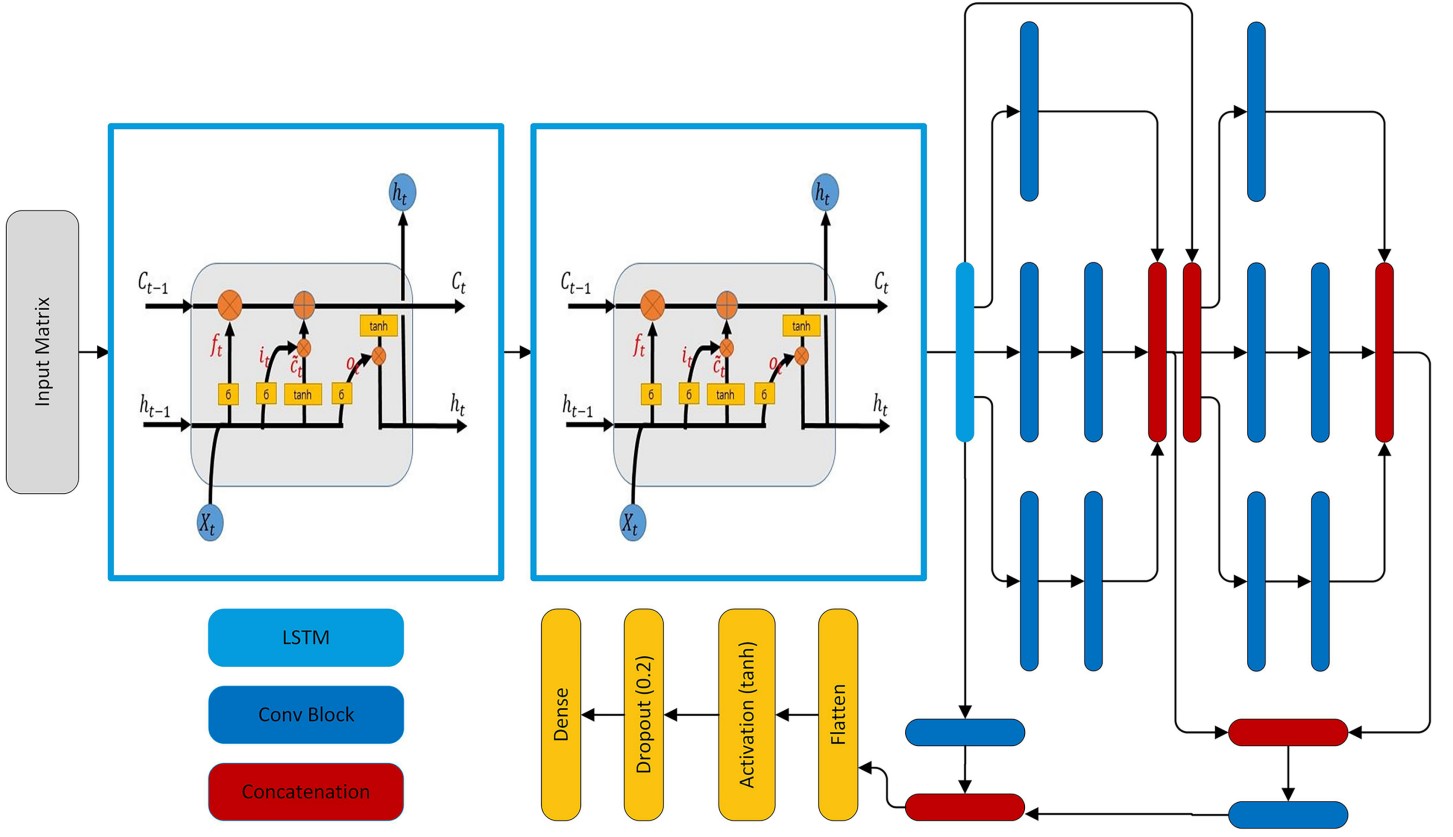

Figure 5  The proposed forecasting model architecture.     

the SC and LSTM extracted features and also increases the hidden layer parameters. So, both paths are followed by a convolutional block having 64 filters with a kernel of size 1. The third skip connection is from the output of the first SC module to the output of the second SC module. The concatenated features are flattened and pass through an activation layer having tanh activation. The activation layer is followed by a dropout layer and finally followed by a dense layer having activation function sigmoid and the number of neurons according to the forecasting interval. Note that in this network convolution layers are used multiple times, so every time the convolution layer is followed by a batch normalization layer and activation function selu, named as a convolutional block shown in Fig. 4A. In addition, L2 regularization with $\lambda = 0.0005$ is used in all convolution layers. The SC module has three parallel paths as shown in Fig. 4B. The first path applies convolution

**Table 7 Basic information of NTDC, AEP, and ISO NE datasets.**

| Dataset name | Start time | End time | Recording interval | Samples |
|---|---|---|---|---|
| NTDC | 2015-01-01 00:00:00 | 2020-04-30 23:00:00 | Hourly | 46,944 |
| AEP | 2004-10-01 01:00:00 | 2018-08-03 00:00:00 | Hourly | 121,296 |
| ISO NE | 2004-01-01 00:00:00 | 2014-12-31 23:00:00 | Hourly | 96,432 |

having a kernel of size 1 which learns local features. The second path has two convolution layers: the first layer with a kernel of size 1 that reduces the dimensions and the second layer has a kernel of size 3. Similar to the second path, the third path also has two convolution layers, in which the first one has a kernel of size one and the second has a kernel of size 5 as shown in Fig. 4B. The hyper-parameters of two inception modules of the model are given in Table 6. The complete architecture is shown in Fig. 5.

## RESULT AND DISCUSSION

The model is developed and trained for the next 24 h load forecasting. All the results and analysis are based on 24-h ahead load forecasting and the same training procedure has been maintained throughout the article unless stated otherwise. Models are developed in tensorflow2.

### Datasets description

The model is trained, debugged, and tested using the NTDC dataset. To validate the generalization capability of the proposed model, the model is evaluated on other two publicly available datasets: AEP (*Mulla, 2018*), a power company under Pennsylvania-New Jersey Maryland (PJM), and ISO-NE (*Chen, 2021*). PJM is a Regional Transmission Organization (RTO). Some preliminary information about these datasets is tabulated in Table 7. All these datasets are split into the train, validation, and test sets in the proportion of 70%, 20%, and 10% respectively.

### Evaluation matrices

In order to evaluate the forecasting performance of the proposed model and its comparison with other deep learning models, three well-known evaluation matrices of time series forecasting are used. These are mean absolute error (MAE), root mean square error (RMSE), and mean absolute percentage error (MAPE). All these matrices calculate the error between the actual and predicted load values, called residual. These matrices are negatively oriented scores, implying that the lower values of these matrices show the betterment of the result. The mathematical expressions are given below (*Shcherbakov et al., 2013*).

$$MAE = \frac{1}{N} \sum_{i=1}^{N} |L_f - L| \qquad (3)$$

$$RMSE = \sqrt{\frac{1}{N}\sum_{i=1}^{N}(L_f - L)^2} \qquad (4)$$

$$MAPE = \frac{1}{N}\sum_{i=1}^{N}|\frac{L_f - L}{L}| \qquad (5)$$

where $N$ is the number of data points in the test dataset, $L$ is the actual value of the load and $L_f$ is the predicted load.

MAE is the mean of the absolute value of the residual which shows that each residual is contributed equally. On the other hand, in RMSE the residual is contributing quadratically or in terms of weighted mean, the weight of each residual is itself which implies a high residual is weighted more. Therefore, RMSE is always greater than MAE and only equal if the residual is uniformly distributed throughout the test dataset. Thus, RMSE is more sensitive to high residuals. On the other hand, MAPE is the mean percentage of absolute relative error, in which the error is divided by the original. Thus, MAPE is not sensitive to the sample values except where the original sample becomes zero, on which the MAPE becomes undefined. In addition, the MAPE is smaller on a symmetrical error, for the one whose actual value is smaller. But MAPE is the percentage equivalent of MAE and thus easily interpretable. Note that the model is trained on normalized load data, so the data needs to be transformed back to actual load values to calculate these parameters.

## Training procedure

The models are built in the tensorflow2 library. Adam optimizer (*Kingma & Ba, 2014*) is used for network training while MAE is used as a loss function. The initial learning rate is 0.001 and batch size is 32. During the training process, callbacks are used to save the model whenever there is an improvement in validation loss. The model which is fine-tuned is either at stagnation range or overfitting point, which is referred to as the best model. The model is further trained and fine-tuned. Fine-tuning is achieved either by reducing the learning rate which decreases the validation loss (*Krizhevsky, Sutskever & Hinton, 2017*) or by increasing the batch size. The batch size increase is also used for tuning with fewer parameter updates (*Smith et al., 2017*).

The validation loss curve has fluctuations at the beginning epochs because of the high initial learning rate and the small batch size shown in Fig. 6. When the validation curve is stagnant and no improvement occurs in the validation loss, or the validation curve deviates from training, the training is stopped. The best model is loaded for calculating MAE, RMSE, and MAPE for all models shown in Table 8. The learning rate is reduced by a factor of 10 and increases the batch size to 256 and starts training. The validation curve fluctuations reduce as shown in Fig. 6. The training is stopped again either by stagnation range or overfitting points. The best model is loaded to calculate MAE, RMSE, and MAPE. It is also pertinent to mention that this method of reducing the learning rate for fine-tuning

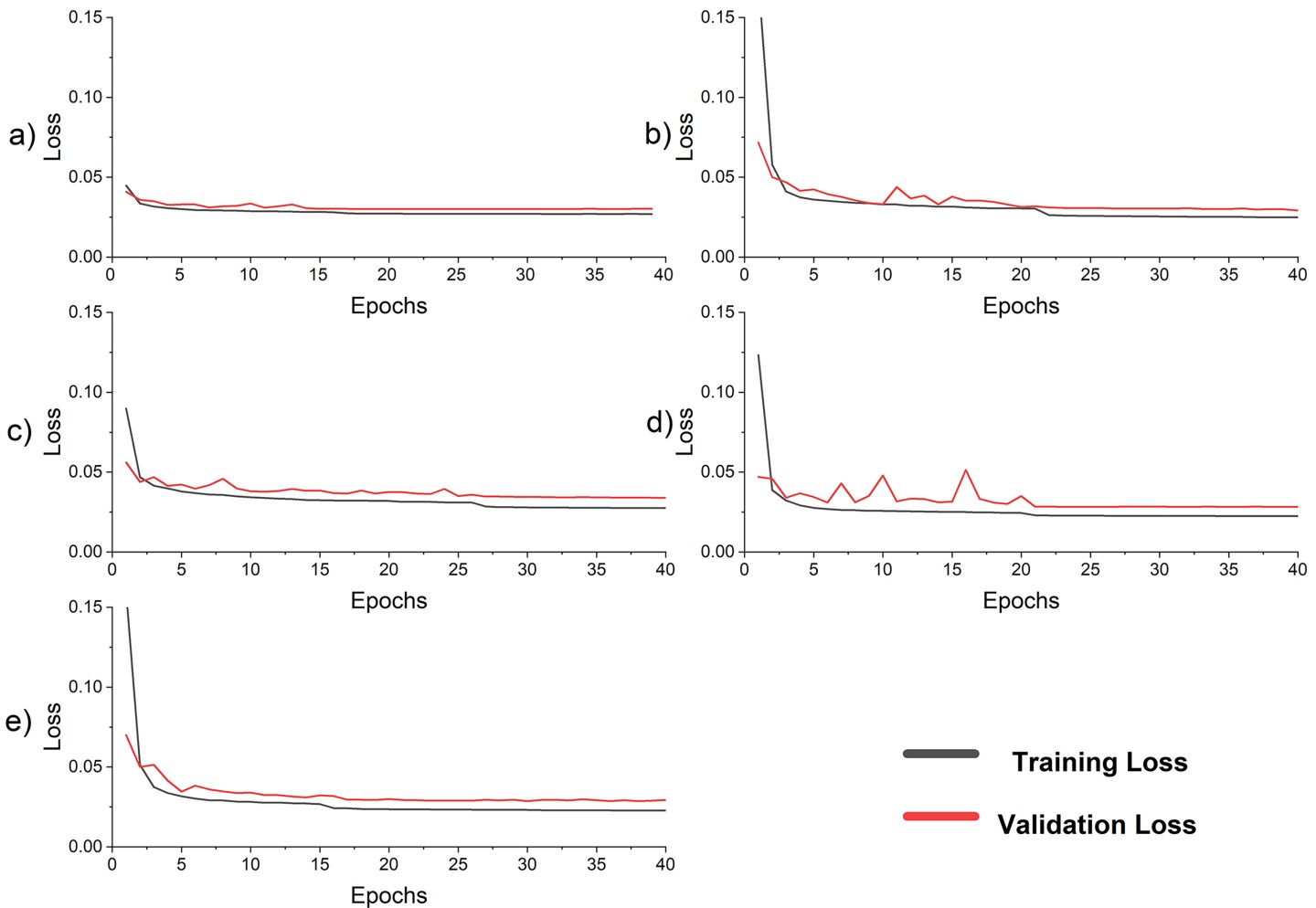

**Figure 6 Loss curve for the models in ablation study (A) LSTM (B) SC (C) SC-LSTM (D) LSTM-SC with skip connection (E) proposed hybrid LSTM-SC.**

**Table 8 Performance evaluation for ablation study.**

| Model | RMSE | MAE | MAPE | % improvement |
|---|---|---|---|---|
| LSTM | 566.66 | 428.79 | 4.26 | 14.82 |
| SC | 517.17 | 388.68 | 3.9 | 5.12 |
| SC LSTM | 619.59 | 483.04 | 4.8 | 29.38 |
| LSTM SC with Skip | 502.14 | 372.53 | 3.72 | 0.27 |
| Proposed LSTM SC | 500.98 | 372.62 | 3.71 | |

gave more fine-grained control to the observer in contrast to the built-in Keras callbacks. In addition, the observer does not know the optimal initial learning rate, the number of epochs at which the learning rate needs to be changed, and the upper and the lower limit on the learning rate.

**Table 9 Performance evaluation parameters at different fine-tuning steps (Ablation study): FE: First best-saved model at epoch and SE: Second best-saved model at epoch.**

| Model | FE | MAPE | SE | MAPE |
|---|---|---|---|---|
| LSTM | 15 | 4.26 | 4 | Not improved |
| SC | 20 | 4.2 | 1 | 3.9 |
| SC LSTM | 25 | 5.29 | 28 | 4.8 |
| LSTM SC with skip | 19 | 4.29 | 5 | 3.72 |
| Proposed LSTM SC | 16 | 3.81 | 29 | 3.71 |

## Ablation study of the proposed model

In this section, the original structure of the model is maintained, some components are removed, and trained the model without them, to show their impact and significance. In addition, LSTM and SC are rearranged in SC LSTM configuration to verify the significance of the LSTM SC configuration. First only the two LSTM layers are trained independently, and the result is tabulated in Table 8. The result shows that the proposed method improves the MAPE on the test dataset by around 14.82%. In the next experiments, only the SC is trained. The improvement of the proposed method is 5.12%, which shows the performance boost of the SC structure. In the following scenario, a hybrid model is developed composed of the cascaded SC and LSTM which further reduces the performance of LSTM by 12.67%. In contrast to the previous case, the LSTM layers are followed by SC layers and the MAPE is 3.71%. Finally, skip connections are introduced in the network, but it has no effect on the performance of the network. This is because the network is not deep and skip connections work well only in deep networks. In the future, this idea will be further exploited. Therefore, the proposed model in the coming experiments is the LSTM-SC. The result of all experiments along with the percentage improvement in MAPE in comparison with the LSTM-SC configuration is tabulated in Table 8. The result shows the significance of the LSTM followed by the SC arrangement. In this arrangement, the LSTM first processes the load consumption data and extracts temporal features. The features are then processed by SC, which extracts hierarchical spatial features. In contrast to the SC-LSTM structure, all the hidden states of LSTM are processed by different size filters which would correct the error in a hidden state, if it occurred at a specific instant. When LSTM directly connects with a dense network, only the final hidden state is processed which may cause an error. Note that all these analyses are based on the same training procedures and on the NTDC dataset. The detail of the sub-experiments is shown in Table 9 and the loss curves for all sub-experiments are shown in Fig. 6. The little fluctuation at the beginning of the loss curve is due to the small batch size of 32 and the high learning rate of 0.001. The MAPE of the proposed model after 16 epochs is 3.81%. It is consistently learning when it is loaded for tuning. The validation curve also closely follows the training curve which is a perfect trade-off between bias and variance.

**Table 10 Multi-step (24-h) forecasting comparison of the proposed model with SOTA on three datasets: NTDC, AEP, and ISO-NE.**

| Model | Model name | Multi-step | | | | | | | | |
| | | AEP | | | NTDC | | | ISONE | | |
| | | RMSE | MAE | MAPE | RMSE | MAE | MAPE | RMSE | MAE | MAPE |
|---|---|---|---|---|---|---|---|---|---|---|
| *Alhussein, Aurangzeb & Haider (2020)* | CNN-LSTM | 824.11 | 636.88 | 4.33 | 752.98 | 585.56 | 6.06 | 764.08 | 565.46 | 3.86 |
| *Kong et al. (2017)* | LSTM | 742.8 | 521.3 | 3.39 | 602.55 | 451.53 | 4.49 | 634.78 | 437.45 | 2.97 |
| *Marino, Amarasinghe & Manic (2016)* | S2S | 1262.83 | 982.4 | 6.66 | 924.76 | 707.69 | 7.31 | 851.78 | 638.78 | 4.36 |
| *Rafi, Deeba & Hossain (2021)* | CNN-LSTM | 718.66 | 536.42 | 3.64 | 603.37 | 438.69 | 4.39 | 659.7 | 466.13 | 3.2 |
| *Ryu, Noh & Kim (2016)* | DNN | 772.3 | 574.57 | 3.87 | 591.9 | 455.75 | 4.55 | 632.05 | 441.24 | 3.01 |
| *Ullah et al. (2019)* | M-BDLSTM | 790.1 | 580.23 | 3.86 | 576.16 | 435.76 | 4.3 | 726.47 | 499.96 | 3.39 |
| *Chen et al. (2023)* | ResNet-LSTM | 735.02 | 539.12 | 3.62 | 600.55 | 459.44 | 4.57 | 649.23 | 450.31 | 3.07 |
| *Ijaz et al. (2022)* | ANN-LSTM | 748.16 | 571.31 | 3.94 | 1,341.3 | 1,017.5 | 7.33 | 734.07 | 535.97 | 3.71 |
| *Hussain et al. (2022)* | Hybrid | 690.38 | 499.28 | 3.33 | 581.68 | 425.35 | 4.29 | 602.43 | 412.92 | 2.84 |
| Proposed | LSTM-SC | 731.3 | 518.81 | 3.46 | 500.98 | 372.62 | 3.72 | 631.93 | 400.67 | 2.7 |

## State-of-the-art methods

In this section, the recent state-of-the-art (SOTA) models (*Ryu, Noh & Kim, 2016*; *Kong et al., 2017*; *Marino, Amarasinghe & Manic, 2016*; *Rafi, Deeba & Hossain, 2021*; *Alhussein, Aurangzeb & Haider, 2020*; *Ullah et al., 2019*; *Ijaz et al., 2022*; *Chen et al., 2023*; *Hussain et al., 2022*; *Shao & Kim, 2020*) are selected from the literature and compared their results with the proposed model. All these models are reproduced by one-to-one correspondence with the source article. Many other relevant articles (*Ding, Liu & Zou, 2021*; *Son et al., 2022*; *Chen et al., 2021*; *Elsaraiti & Merabet, 2022*) are not reproducible in their true spirit because their many hyper-parameters are not listed. Research article (*Ryu, Noh & Kim, 2016*) used a deep neural network, (*Kong et al., 2017*) used LSTM, and (*Marino, Amarasinghe & Manic, 2016*) used sequence to sequence. On the other hand (*Shao & Kim, 2020*) used three parallel channels of LSTM and K-means classifier. Similarly, (*Rafi, Deeba & Hossain, 2021*; *Alhussein, Aurangzeb & Haider, 2020*; *Ullah et al., 2019*; *Ijaz et al., 2022*; *Chen et al., 2023*; *Hussain et al., 2022*) used hybrid models composed of CNN and RNN variants for load forecasting.

## 24-h forecasting comparative analysis

The proposed model is used to forecast the coming 24 h load consumption. The model is trained, validated, and tested on all three datasets with the given number of samples as in Table 7. The results of the proposed and all SOTA models are shown in Table 10. The result indicates that the proposed model outperforms all the comparative models on all three datasets. In all cases, RMSE is greater than MAE because the residual is not uniformly distributed and the test data has outliers, as the test data include last month's data which has outliers as shown in Fig. 2D. In addition, the features incorporated along with load consumption have a strong correlation with load consumption. To investigate their effect on MAPE, these features are removed from the NTDC dataset and retrained the model only on the load data. The MAPE is increased to 3.96% which clearly indicates the

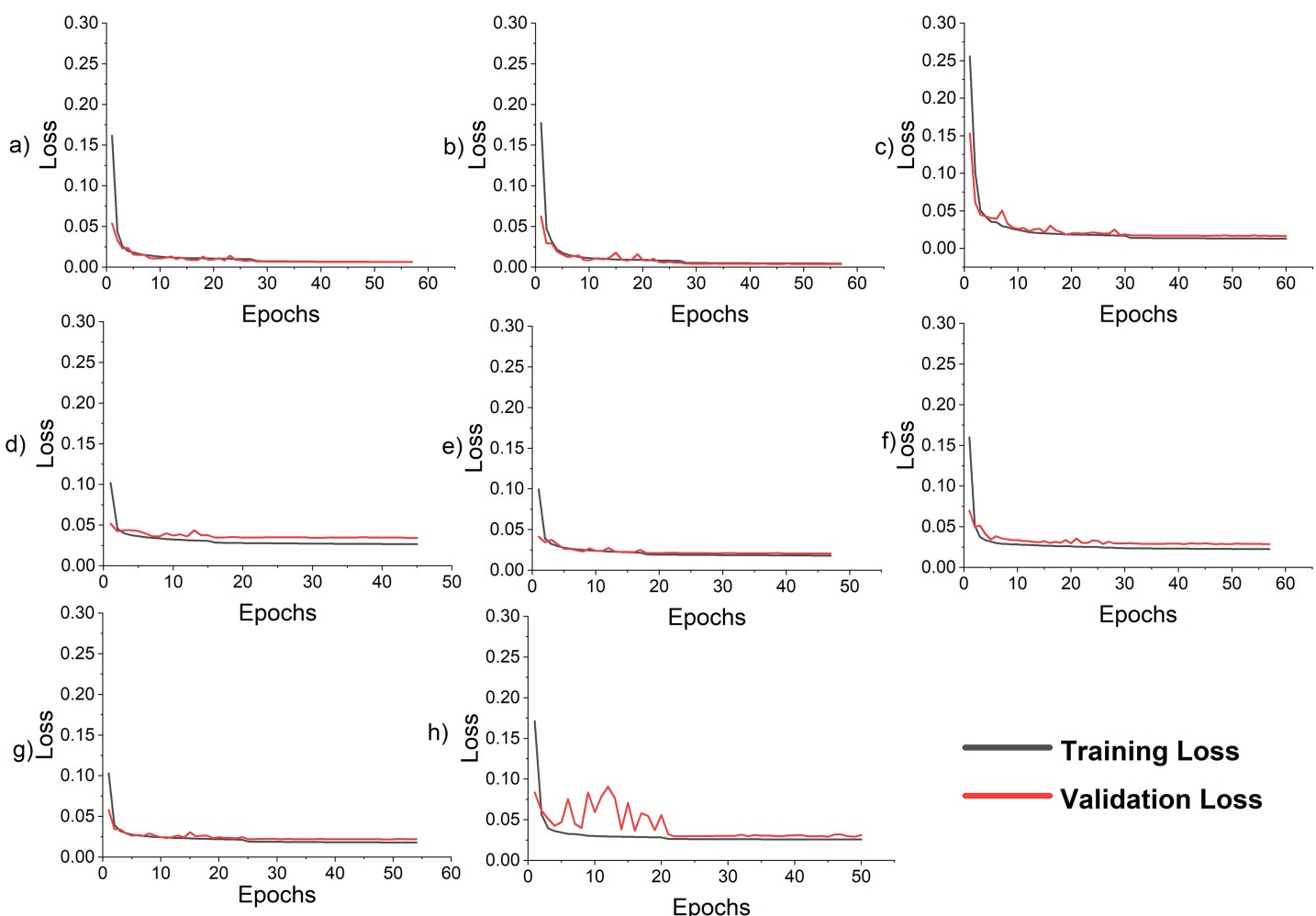

**Figure 7** Loss curve for the LSTM-SC model for single-step and multi-step forecasting (A) single-step AEP (B) single-step ISO-NE (C) single-step NTDC (D) multi-step AEP (E) multi-step ISO-NE (F) multi-step NTDC (G) multi-step ISO-NE without temperature (H) multi-step ISO-NE without features.

effectiveness of the proposed preprocessing. The corresponding loss curve is shown in Fig. 7H. The curve fluctuates more at the beginning because the model is not learning most of the load features without calendric features at batch size 32 and a learning rate of 0.001. After the learning rate reduction and batch size increase the model starts learning and the model improves. Furthermore, the temperature present in ISO-NE datasets has a strong correlation with load consumption. When it is removed, the model is trained and tested again while keeping all the conditions remains the same. The MAPE has increased to 2.9% which is a 7.4% increase. As the temperature is a highly correlated feature that varies differently from previous historical data and depends on changing environments with time. So, a small noise in temperature data can affect the forecast very much. For this, the effect of randomness is investigated by modifying the actual temperature date by adding Gaussian noise with a standard deviation of 1, 2, and 3 degrees. This increases the MAPE

**Table 11 The performance evaluation of the proposed model and SOTA models on 12 step ahead load forecasting on AEP datasets.**

| Dataset | Reference | Model name | MAE | RMSE | MAPE |
|---------|-----------|-----------|------|------|------|
| AEP | *Shao & Kim (2020)* | TL-MCLSTM | 430.9 | 551.8 | 3.1 |
| | *Xue et al. (2019)* | XGBoost | 718.0 | 551.8 | 5.2 |
| | *Kong et al. (2017)* | LSTM | 434.4 | 560.0 | 3.1 |
| | *Yan et al. (2018)* | Hybrid | 646.3 | 802.9 | 4.84 |
| | *Shao, Kim & Sontakke (2020)* | CNN-LSTM | 510.6 | 646.2 | 3.7 |
| | Proposed | LSTM-SC | 386.17 | 549.92 | 2.58 |

from 2.70 to 2.73, 2.80, and 2.91 respectively. This clearly indicates that the proposed model performance varies a little bit due to temperature variation. The Loss curves for temperature and without temperature case is shown in Figs. 7E and 7H respectively. The curves are similar but the curve for the temperature inclusion case comes a little down. The loss curves for AEP, ISO-NE, and NTDC are shown in Figs. 7D–7F respectively. The Loss curve of NTDC data set is a little bit higher than the other two dataset because of the small amount of data in NTDC dataset.

## 12-h forecasting comparative analysis

The performance of proposed model is also compared with recent state-of-the-art model (*Shao & Kim, 2020*) for forecasting next 12 h load consumption. The model was trained on multiple datasets but for comparison purposes, only AEP is chosen. The results are compared with Table 3 of the article. The summarized results are tabulated in Table 11. The result shows the reduction in forecasting error.

## Single-step forecasting comparative analysis

The actual power consumption and the predicted power of the proposed model and all other state-of-the-art models are plotted in Fig. 8. The prediction is done for the complete test dataset, but a few days from the beginning are drawn to be more visible. In addition, the performance evaluation matrices are tabulated in Table 12. These results show the effectiveness of the proposed hybrid model. The loss curves of the proposed model on AEP, ISO-NE, and NTDC datasets are shown in Figs. 7A–7C respectively. These curves are asymptotic on the epoch axis which shows that single-step forecasting is an easy task in comparison to the multiple-step ahead task. In addition, in the case of ISO-NE, the curves touch the epoch axis because the data includes temperature, an extra input variable, and it helps the model to learn more.

## LIMITATIONS AND FUTURE WORK

The specialized proposed architecture LSTM-SC which is the combination of special neural network modules enhanced the performance of STLF. Similarly, there are many other recent deep neural network architectures, specialized for different data types and applications. Therefore, by carefully analyzing the nature of the data and selecting an

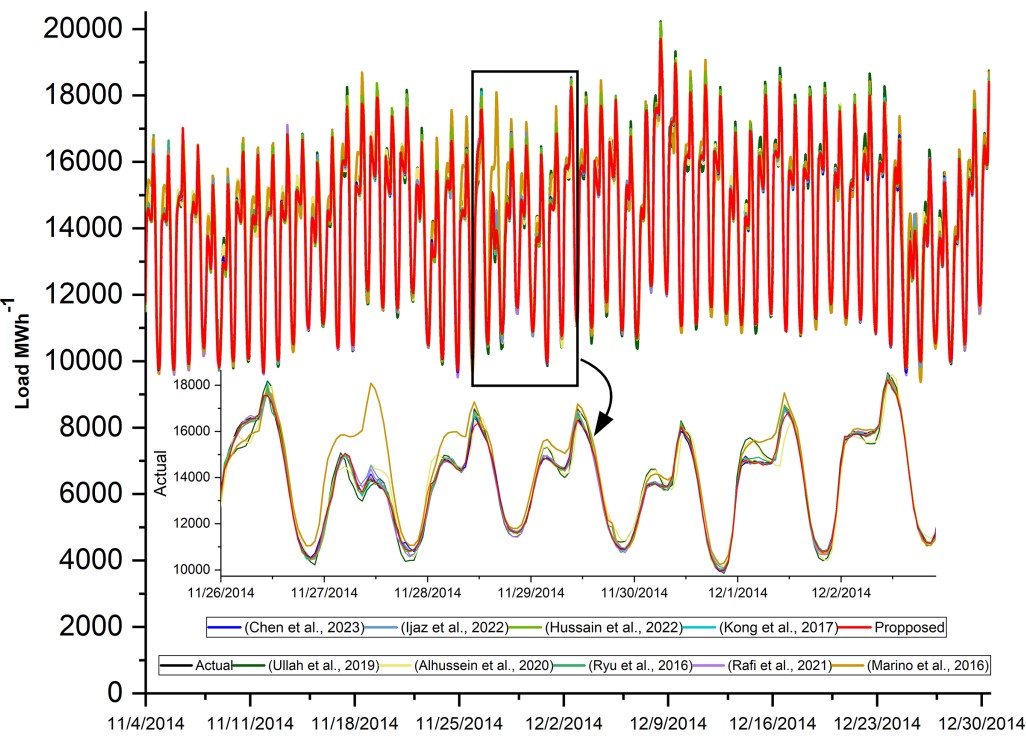

**Figure 8 Actual and predicted load consumption of LSTM-SC and SOTA models.**

**Table 12 Single-step forecasting comparison of the proposed model with SOTA on three datasets: NTDC, AEP, and ISO-NE.**

| Model | Model name | Single-step | | | | | | | | |
|---|---|---|---|---|---|---|---|---|---|---|
| | | AEP | | | NTDC | | | ISONE | | |
| | | RMSE | MAE | MAPE | RMSE | MAE | MAPE | RMSE | MAE | MAPE |
| *Alhussein, Aurangzeb & Haider (2020)* | CNN-LSTM | 381.66 | 305.71 | 2.15 | 570.66 | 439.79 | 4.49 | 394.26 | 289.62 | 2.01 |
| *Kong et al. (2017)* | LSTM | 133.25 | 97.36 | 0.66 | 368.43 | 277.54 | 2.74 | 132.17 | 69.73 | 0.48 |
| *Marino, Amarasinghe & Manic (2016)* | S2S | 975 | 758.58 | 5.11 | 925.16 | 734.03 | 7.19 | 571.82 | 403.24 | 2.76 |
| *Rafi, Deeba & Hossain (2021)* | CNN-LSTM | 153.65 | 118.06 | 0.82 | 351.63 | 271.56 | 2.69 | 161.3 | 98.26 | 0.7 |
| *Ryu, Noh & Kim (2016)* | DNN | 140.34 | 107.68 | 0.74 | 359.89 | 275.66 | 2.72 | 138.13 | 74.41 | 0.52 |
| *Ullah et al. (2019)* | M-BDLSTM | 355.42 | 271.86 | 1.83 | 569.19 | 436.18 | 4.36 | 323.78 | 229.61 | 1.57 |
| *Chen et al. (2023)* | ResNet-LSTM | 332.35 | 249.08 | 2.48 | 310.97 | 234.68 | 2.32 | 140.7 | 78.39 | 0.54 |
| *Ijaz et al. (2022)* | ANN-LSTM | 157.2 | 115.54 | 0.8 | 877.03 | 643.37 | 4.54 | 175.98 | 93.37 | 0.65 |
| *Hussain et al. (2022)* | Hybrid | 148.29 | 110.24 | 0.75 | 267.46 | 355.84 | 2.62 | 139.98 | 73.7 | 0.51 |
| Proposed | LSTM-SC | 130.08 | 97.78 | 0.67 | 322.99 | 244.22 | 2.38 | 140.03 | 67.59 | 0.47 |

appropriate model for each type of data, and then appropriately connecting them in a hybrid model may improves the performance. This combination may be either in parallel or series or any other combination.

For proper energy management, for the smooth operation of smart grid needs an optimal forecasting algorithm which helps in decision making. In the future, the proposed

model can be tested in combination with energy management algorithms like reinforcement learning in some energy management applications.

The proposed model forecasts a single value. However, future uncertainty can best be described by a range of possible values or a distribution. The cause of this uncertainty is either due to the stochastic nature of deep learning models or due to the uncertainty in the input matrix. The model uncertainty is due to different reasons like parameter initialization and updating them in a stochastic way. On the other hand, the uncertainty in other input co-related quantities like temperature is due to noise or variation with time. However, most of the dataset have not included all co-related features and are usually small in size. Therefore, in the future, a large dataset with all co-related features is required. Furthermore, the methodology of the proposed work can be extended to include such stochastic variation in both model and data.

Smart grid operation required the forecasting of different complex varying quantities like load demand, generation by intermittent renewable energy resources, different market situations, and weather or environmental conditions. Therefore, a single forecasting algorithm that can forecast all these quantities becomes a need of the hour. The proposed model is a multi-horizon on three different regional datasets. In the future, the proposed model can be trained and tested for different data and may also used for transfer learning.

## CONCLUSION

This article addresses the issue of multi-horizon short-term load forecasting in a more precise way. This article proposed a novel hybrid method with the integration of LSTM and a modified split-convolution network. The results show that the preceding LSTM from CNN and further modification of CNN by splitting the CNN into parallel paths, each having kernel of different sizes for extracting both local and global features. Furthermore, it also processes all hidden states instead of the last hidden state, which improves performance. The performance of the model is evaluated on the Pakistan National Power dataset and two other publicly available datasets: AEP and ISO-NE. Further, it is also investigated that combining multiple correlated features with load data improves the performance of the proposed network which shows that the network is effectively designed for extracting features from the data. In addition, random Gaussian noise with different standard deviations is added to temperature in the ISO-NE dataset and its effects are investigated. Comparing the results of the proposed model with other state-of-art models on different publicly available datasets indicated that it has a strong generalization capability and less error in forecasting.

### Funding
This research is funded by the Researches Supporting Project (RSPD2023R947), King Saud University, Riyadh, Saudi Arabia. The funders had no role in study design, data collection and analysis, decision to publish, or preparation of the manuscript.

## Grant Disclosures

The following grant information was disclosed by the authors:
King Saud University, Riyadh, Saudi Arabia: RSPD2023R947.

## Competing Interests

Khursheed Aurangzeb is an Academic Editor for PeerJ.

## Author Contributions

- Irshad Ullah conceived and designed the experiments, performed the experiments, analyzed the data, performed the computation work, prepared figures and/or tables, and approved the final draft.
- Syed Muhammad Hasanat conceived and designed the experiments, performed the experiments, performed the computation work, prepared figures and/or tables, and approved the final draft.
- Khursheed Aurangzeb conceived and designed the experiments, performed the experiments, analyzed the data, authored or reviewed drafts of the article, and approved the final draft.
- Musaed Alhussein analyzed the data, authored or reviewed drafts of the article, and approved the final draft.
- Muhammad Rizwan analyzed the data, authored or reviewed drafts of the article, and approved the final draft.
- Muhammad Shahid Anwar analyzed the data, authored or reviewed drafts of the article, and approved the final draft.

## Data Availability

The data and code are available in the Supplemental Files.

The code is available at GitHub and Zenodo: https://github.com/SyedHasnat/Papers/tree/main/PeerJ.

Syed Muahammad, Hasanat, Irshad, Ullah, Khursheed Aurangzeb, Musaed Alhussein, Muhammad Rizwan, & Muhammad Shahid Anwar. (2023). Multi-Horizon Short-Term Load Forecasting Using Hybrid of LSTM and Modified Split Convolution (1.1.1). Zenodo. https://doi.org/10.5281/zenodo.8060005.

## Supplemental Information

Supplemental information for this article can be found online at http://dx.doi.org/10.7717/peerj-cs.1487#supplemental-information.

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
