# Peer review of "Multi-horizon short-term load forecasting using hybrid of LSTM and modified split convolution"

_PeerJ Computer Science, doi:10.7717/peerj-cs.1487_

## Round 0.1 · original submission · Major Revisions

I have received reviews of your manuscript from three scholars who are experts on the cited topic. They find the topic very interesting; however, several concerns must be addressed regarding the organization, experimental results, and comparisons with current approaches. These issues require a major revision. Please refer to the reviewers’ comments listed at the end of this letter, and you will see that they are advising that you revise your manuscript. If you are prepared to undertake the work required, I would be pleased to reconsider my decision. Please submit a list of changes or a rebuttal against each point that is being raised when you submit your revised manuscript.

Thank you for considering PeerJ Computer Science for the publication of your research.

Reviewer 1 ·

Basic reporting

This article proposes a novel hybrid technique based on long short-term memory (LSTM) and a modified Split-Convolution (SC) neural network (LSTM-SC) is proposed for single-step and multi-step
STLF.

Experimental design

There are few observations/suggestions for the authors.

1. Introduction section need to be separated from the related work. Authors have mentioned the contributions. The literature gap needs to be highlighted.

2. Data cleaning could be a part of methodology. Authors can shift this part to the methodology section.

3. Figure 3 must be modified as an architecture/block diagram which signifies each main module.

4. Experimental verification needs to be reframed as results and discussion.

Validity of the findings

This article proposes a novel hybrid technique based on long short-term memory (LSTM) and a modified Split-Convolution (SC) neural network (LSTM-SC) is proposed for single-step and multi-step
STLF. The paper is well written and may be accepted for publication after incorporating the suggestions mentioned.

Reviewer 2 ·

Basic reporting

The English language used needs to be improved to make the manuscript easier to follow.

The authors could add a Related Work Section, to present the background and other significant works in the field.

The structure of the paper conforms to PeerJ standards.

Figures are labbeled correctly and in high-quality, with the only exception of Figure 2.

Raw data are not shared.

Experimental design

The original primary research is within Scope of the journal. The method is well described with sufficient detail.

Validity of the findings

The results present an improvement with respect to state-of-the-art results. The authors could highlight also the limitations of the proposed approach.

Additional comments

Minor notes:
- The labels of Figure 2 are stretched
- The text of Table 5 is too small
- A proofread is needed to correct errors and typos

Reviewer 3 ·

Basic reporting

no comment

Experimental design

no comment

Validity of the findings

no comment

Additional comments

In this research, a novel hybrid technique based on long short-term memory (LSTM) and a modified Split-Convolution (SC) neural network (LSTM-SC) is proposed for single-step and multi-step STLF. The concatenating order of LSTM and SC in the proposed hybrid network provides an excellent capability of extraction of sequence-dependent features and other hierarchical spatial features. The model is evaluated by the Pakistan National Grid load dataset recorded by the National Transmission and Dispatch Company (NTDC). The load data is pre-processed and multiple other correlated features are incorporated into the data for performance enhancement. For generalization capability, the performance of LSTM-SC is evaluated on publicly available datasets of American Electric Power (AEP) and Independent System Operator New England (ISO-NE). The effect of temperature, a highly correlated input feature, on load forecasting is investigated either by removing the temperature or adding a Gaussian random noise into it. The proposed method has less forecasting error,
strong generalization capability, and satisfactory performance on multi-horizon. The concept of the paper is good. My comments are given below:
1. The motivation of the research is missing in the Introduction section. Also, highlight the novelty of the study in the Introduction section of the paper.
2. The literature survey is weak; the authors must present a comprehensive analysis of the literature and how it has led to gaps that the work tries to address. Without a strong literature survey, the authors' work may lack context and relevance to the field. A comprehensive analysis of the literature would provide a foundation for understanding the gaps in knowledge that the authors aim to fill with their research.
3. The authors must proofread the manuscript for various ill framed sentences and typos.
4. All variables in the equations should be defined. For example, the variables in Equation 2 are not defined.
5. The quality of figures should be improved.
6. The performance of the proposed system should be compared with more recent related papers.
7. The author should be discussed the threats to validity of the proposed method.
8. Introduce a new section named future enhancement and place it with sufficient contents before the conclusion section.
9. The limitation of the current study should be discussed.

---

## Round 0.2 · accepted · Accept

I am pleased to inform you that your work has now been accepted for publication in PeerJ Computer Science.

Please be advised that you are not permitted to add or remove authors or references post-acceptance, regardless of the reviewers' request(s).

Thank you for submitting your work to this journal. On behalf of the Editors of PeerJ Computer Science, we look forward to your continued contributions to the Journal.

With kind regards,

Reviewer 1 ·

Basic reporting

Authors have incorporated all the suggestions provided in the review. A small correction needs to be done. The git hub link of the code given in abstract section could be shifted to last line of results section.

Experimental design

Authors have incorporated all the suggestions provided in the review. A small correction needs to be done. The git hub link of the code given in abstract section could be shifted to last line of results section.

Validity of the findings

NA

Additional comments

NA

Reviewer 2 ·

Basic reporting

No comment

Experimental design

No comment

Validity of the findings

No comment

Additional comments

The authors well addressed all the raised issues